# Identifying efficient linkage strategies for men (IDEaL): a study protocol for an individually randomised control trial

Kathryn Dovel ![ORCID] ,[1,2] Kelvin Balakasi,[2] Julie Hubbard,[1] Khumbo Phiri,[2] Brooke E Nichols ![ORCID] ,[3,4,5] Thomas J Coates,[1,6] Michal Kulich,[7] Elijah Chikuse,[2] Sam Phiri,[2] Lawrence C Long,[3,8] Risa M Hoffman,[1] Augustine T Choko[9,10]

For numbered affiliations see end of article.

**Correspondence to**
Dr Kathryn Dovel;
kdovel@mednet.ucla.edu

## ABSTRACT

**Introduction** Men in sub-Saharan Africa are less likely than women to initiate antiretroviral therapy (ART) and more likely to have longer cycles of disengagement from ART programmes. Treatment interventions that meet the unique needs of men are needed, but they must be scalable. We will test the impact of various interventions on 6-month retention in ART programmes among men living with HIV who are not currently engaged in care (never initiated ART and ART clients with treatment interruption).

**Methods and analysis** We will conduct a programmatic, individually randomised, non-blinded, controlled trial. 'Non-engaged' men will be randomised 1:1:1 to either a low-intensity, high-intensity or stepped arm. The low-intensity intervention includes one-time male-specific counseling+facility navigation only. The high-intensity intervention offers immediate outside-facility ART initiation+male-specific counselling+facility navigation for follow-up ART visits. In the stepped arm, intervention activities build in intensity over time for those who do not re-engage in care with the following steps: (1) one-time male-specific counselling+facility navigation→(2) ongoing male mentorship+facility navigation→(3) outside-facility ART initiation+male-specific counselling+facility navigation for follow-up ART visits. Our primary outcome is 6-month retention in care. Secondary outcomes include cost-effectiveness and rates of adverse events. The primary analysis will be intention to treat with all eligible men in the denominator and all men retained in care at 6 months in the numerator. The proportions achieving the primary outcome will be compared with a risk ratio, corresponding 95% CI and p value computed using binomial regression accounting for clustering at facility level.

**Ethics and dissemination** The Institutional Review Board of the University of California, Los Angeles and the National Health Sciences Research Council in Malawi have approved the trial protocol. Findings will be disseminated rapidly in national and international forums and in peer-reviewed journals and are expected to provide urgently needed information to other countries and donors.

**Trial registration number** NCT05137210.
**Date and version** 5 May 2023; version 3.

## STRENGTHS AND LIMITATIONS OF THIS STUDY

⇒ Identifying efficient linkage strategies for men (IDEaL) provides male-specific differentiated models of care aimed to improve men's antiretroviral therapy outcomes. We specifically focus on building trusting relationships with healthcare workers and developing client-led, individualised strategies to overcome barriers to care.

⇒ IDEaL will test the impact of a stepped intervention for men. This approach promises to improve the efficiency and reach of HIV programmes for men as the highest-resource interventions will only be received by the minority of men who are most in need.

⇒ IDEaL develops and tests a male-specific counselling curriculum that, if effective, could easily be taken to scale. Findings from the study will identify critical components for male-specific counselling, especially among men who struggle to be retained in HIV care.

⇒ IDEaL interventions do not change facility characteristics that may act as barriers to men's use of facility-based services. IDEaL focuses on providing outside-facility services for reaching men.

## INTRODUCTION

Men in sub-Saharan Africa (SSA) are underrepresented in HIV programmes.[1] Men are less likely than women to know their HIV status and to initiate antiretroviral therapy (ART), and more likely to face treatment interruptions once in care.[2] Only 69% of men who start ART reach viral suppression compared with 77% of women.[2] As a result, men in the region are 37% more likely to die from AIDS-related causes as compared with women.[3]

One contributor to men's poor HIV outcomes is an increased risk of disengagement from care. Engagement in ART programmes is not static—many ART clients cycle through care, starting and stopping HIV care multiple times throughout their lifetime.[4 5] Up to 46% of ART clients experience treatment interruption,[6–8] and between 30%

and 40% of those who return experience repeat treatment interruption within 6 months.[9 10] Men are particularly prone to cycling through ART programmes, with more frequent stop–start instances and longer periods outside of care as compared with women.[6 11–13] Improving men's long-term engagement in HIV care is critical for men's health and reducing HIV transmission.[14]

Men who disengage from HIV programmes (either after testing HIV positive or after enrolling in HIV care services) are frequently described as a difficult and 'hard-to-reach' population.[15 16] However, growing evidence suggests that men desire HIV services[17 18] but encounter multiple health systems barriers to care that make it impossible to stay in long-term care.[19] There is an urgent need to develop client-centred strategies tailored to men that facilitate men's engagement and re-engagement in HIV treatment programmes.

Some men may require male-specific interventions to facilitate engagement in HIV care. Men have less exposure to HIV services than women[19 20] and work demands may conflict with ART clinic schedules.[21 22] Difficult interactions with healthcare workers (HCWs) can also prevent men from engaging or re-engaging in care.[23 24] Furthermore, most ART counselling curricula do not target men and often lack the client-centred counselling needed to develop internal motivation to engage and stay engaged in care.

Differentiated service delivery (DSD) models are now being developed to improve men's ART engagement throughout SSA.[25–27] As DSD models for men are developed, it is critical that strategies be feasible and cost-effective to allow scale-up. A 'one-size-fits-all' model is not as effective as more nuanced approaches.[28–30] Stepped interventions increase in intensity over time and are purposely designed to address prevailing barriers in the target population in order to positively affect the desired outcome.[31 32] An incremental, stepped approach may be the most appropriate and scalable way to improve men's care in low-resource settings. Men are not homogeneous: some men may require minimal support to engage in care, while others may require extensive support. Stepped interventions allow programmes to target the highest-resource interventions to the minority of men who need them most.

The *Identifying efficient linkage strategies for men (IDEaL) trial* is an individually randomised control trial aimed to test the impact of various interventions on ART (re)initiation and 6-month retention among men living with HIV who are not currently engaged in HIV care in Malawi. We will compare a stepped intervention against low-intensity and high-intensity interventions to assess the impact of the stepped intervention on men's use of ART services over time (see online supplemental file 1). The trial contributes to existing literature by testing male-specific, client-centred strategies to re-engage men in care. This is one of the first trials specifically designed with men's re-engagement in care in mind. If effective, such interventions may decrease repeat treatment interruption and duration of treatment interruptions among men, which can improve viral suppression and reduce onward HIV transmission.[14]

## METHODS AND ANALYSIS
### Objectives
Our primary objective is to test the effect of a male-specific, stepped intervention on men's 6-month retention in ART care compared with male-specific low-intensity and high-intensity interventions (retention is defined as <28 days late for their 6-month ART appointment). Secondary objectives are to understand the effect of a stepped intervention on: (1) ART initiation; (2) the presence of adverse events (ie, unwanted disclosure, end of relationship or intimate partner violence (IPV)); (3) intervention acceptability and (4) cost-effectiveness.

### Trial design
IDEaL is a programmatic, individually randomised, non-blinded controlled trial design. We will recruit men from 13 high-burden health facilities in Malawi using medical chart reviews to identify men who are living with HIV but not engaged in HIV care.

### Randomisation
Individual men will be block randomised using R by a biostatistician using a 1:1:1 ratio to either the stepped, low-intensity or high-intensity study arm using a computer-generated program. Participants will be randomised in blocks of 3 and 6, depending on the number of men available for recruitment at each facility. After enrolling in the trial and completing a baseline survey, men will be assigned to a study ID based on the randomisation list. Study IDs will be linked with the preassigned blocked randomisation and preloaded into the tablet device, but will be unknown to the study staff until survey and randomisation modules are completed and saved, ensuring randomisation cannot be manipulated by the study staff. Once finalised, the randomisation results will appear on the tablet device as a picture, and will be shown to the participant to maximise transparency and study buy-in.

### Interventions
The effectiveness of the stepped intervention will be compared with a low-intensity intervention (one time male-specific counseling+facility navigation defined as escort to the facility (if desired) and orientation to the ART clinic and procedures) and with a high-intensity intervention (outside-facility ART initiation+male-specific counselling+facility navigation for follow-up ART visits). Across all arms, men who do not (re)initiate ART will continue receiving follow-up visits for up to 3 months, depending on preferences of the client. The number of intervention visits delivered for each participant will be documented.

### Arm 1: low-intensity arm: male-specific counseling+facility navigation
Participants randomised to the low-intensity arm will be traced in the community and receive a one-time,

one-on-one male-specific counselling session, using client-centred service techniques.[33] All counselling sessions will be completed by a lay cadre male HIV counsellor (called Treatment Supporter in Malawi) trained in the study counselling curriculum. Treatment Supporters are responsible for routine tracing, linkage support medical record documentation and counselling.

The male-specific curriculum is developed specifically for this trial. Ministry of Health counselling materials are adapted to meet the specific needs of men, based on formative in-depth interviews, focus group discussions and a systematic literature review. Adaptations will include exploring topics of most concern to men in Malawi (ie, earning money while HIV positive, side effects and concerns regarding lifelong medication, ART as a tool to provide and care for family, etc). The materials will also include language and pictures that resonate with men (ie, emphasising how HIV and HIV services interact with men's strength, responsibility, planning for the future), and male-specific case studies of challenges men face and how they overcome them. The adapted male-specific counselling curriculum will be developed into a standardised counselling flip chart (ie, job aid).

Men who wish to (re)initiate ART will be offered facility navigation and facility-based services. Participants will be escorted to the facility (if desired), orientated to the ART clinic procedures and introduced to other HCWs who routinely work at the facility. Facility navigation is intended to facilitate a positive experience for men by helping them feel comfortable and confident navigating clinic spaces. Men may access all ART clinical services at the health facility (but counselling described above will be provided in the community). The client will be responsible for all transport costs related to return to the facility in all study arms.

Participants who do not (re)initiate care within 14 days will be offered follow-up counselling every 2 weeks (up to 3 times) until participants (re)initiate care or inform the counsellor that they do not wish to be contacted.

## Arm 2: high-intensity arm: male-specific counselling+outside-facility ART initiation+facility navigation

Participants in the high-intensity arm will be offered community-based male-specific counselling by HIV counsellors (described above), and offered ART (re)initiation outside facility (either at home or another location in the community of their choice). Those who choose outside-facility ART initiation will be referred to a male study nurse who will meet participants one on one at times and locations that are convenient for participants. The nurse will offer a brief counselling session, reviewing key topics from male-specific counselling curriculum that are most relevant to the individual participant. Nurses will then conduct WHO staging. Individuals classified as WHO stage 3 or 4 will be referred (and escorted, if desired) to the nearest public health facility for additional services. Participants classified as WHO stage 1 or 2 will be given same-day ART. At their 4-week follow-up ART appointment, participants will receive facility navigation by the same nurse and afterward access ART services.

Participants who choose facility-based ART (re)initiation will be referred (and escorted, if desired) by the HIV counsellor (lay cadre) to the nearest facility of their choice and receive facility navigation on a day and time that is convenient for them.

Men who do not (re)initiate care will be offered biweekly follow-up counselling at times and intervals determined by participants' preferences until they engage in care or inform the nurse that they no longer wish to be contacted.

## Arm 3: stepped arm

The stepped arm will build in intensity over time for those who have not (re)initiated care 14 days after study enrolment, or who do not return for their first ART follow-up appointment after (re)initiation (see figure 1).

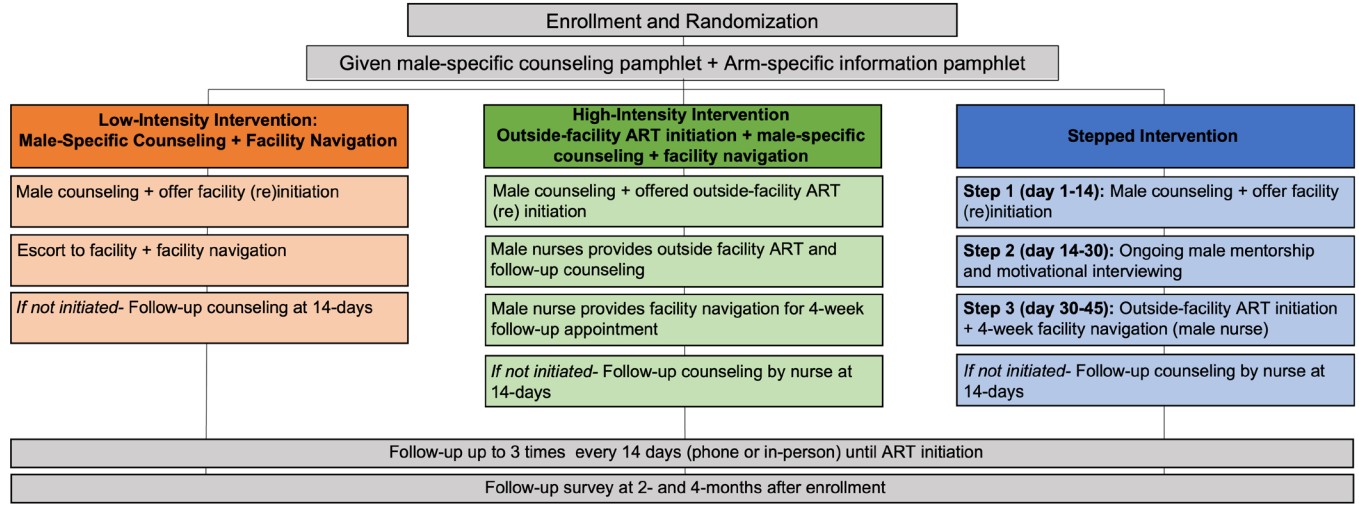

**Figure 1** Trial design. ART, antiretroviral therapy.

Individuals will move to the next 'step' every 2 weeks, moving from the lightest to the most intensive interventions over the course of 6 weeks until (re)initiation has been achieved or they inform the counsellor they do not wish to be contacted. The stepped arm includes the following steps:

### Step 1: male-specific counselling+facility navigation and facility-based ART services

Step 1 includes the same components described in the low-intensity arm. Briefly, participants will be traced in the community and receive a one-time male-specific counselling session. Men who wish to (re)initiate care will be provided facility navigation and standard of care facility-based services.

### Step 2: ongoing motivational interviewing+facility navigation and facility-based ART services

Men who do not (re)initiate care (either have not engaged ART within 14 days of enrolment or do (re)initiate ART but are >7 days late for a follow-up appointment) will move to the next 'step' of the intervention, which adds ongoing motivational interviewing to their package of activities. Motivational interviewing is a client-centred, client-led method for counselling that helps participants identify barriers to a desired outcome and develop personalised solutions.[34 35] The strategy has successfully been used with ART clients.[36] Mentors will work with participants to: (1) build self-efficacy, (2) identify internal motivations for the desired behaviour, and (3) establish strategies and short-term and long-term goals needed to reach ART initiation and retention. A male mentor specifically trained in motivational interviewing adapted to the local context and male population will provide ongoing, one-on-one in-depth counselling, motivational interviewing and general 'check-ins' approximately twice within a 2-week period. The mentor will not necessarily be HIV positive (unlike other mentorship models) as the Malawi Ministry of Health has moved away from HIV-positive peer mentor cadres. However, they will be experienced in HIV counselling and trained in male-specific needs. Motivational interviewing will take place in a location preferred by the participant, likely in the community. Participants who choose to (re)initiate ART can access ART services at the facility of their choice and will be given facility navigation as described in arm 1.

### Step 3: outside-facility ART initiation+male-specific counselling

Men who are not engaged after step 2 (either have not (re)initiated ART <14 days after moving to step 2 or did (re)initiate but are >7 days late for a follow-up ART appointment) will be offered outside-facility ART by a male nurse certified in HIV counselling. Steps will follow those outlined in the high-intensity arm, with a brief counselling session, WHO staging, same-day ART reinitiation for those WHO stage 1 or 2, and facility navigation for their 4-week follow-up appointment.

### Trial setting

The study will take place in central and southern Malawi. Malawi has an HIV prevalence of 9.6%,[37] and of the estimated 330 000 men living with HIV in the country, 54 500 are not in care.[38] Men in Malawi live in primarily rural settings, are self-employed and subsistence farmers; the minority have regular access to a private phone and most are highly mobile.[39 40]

### Population

We will recruit men from 13 high-burden health facilities in Malawi, using medical chart reviews to identify men living with HIV who are not engaged in HIV care. Study facilities will vary by facility type (hospital/health centre), management (public/mission), location (rural/urban) and region (central/southern Malawi).

Eligibility criteria for men include: (1) ≥15 years of age; (2) live in facility catchment area; and (3) tested HIV positive and either (a) self-report having not yet initiated ART within 7 days of testing HIV positive, (b) initiated ART but are at risk of immediate default (ie, ≥7 days late for their 4-week ART refill appointment), or (c) initiated ART and attended their first refill appointment but later defaulted (ie, >28 days late to care). For those who never initiated ART and do not have proof of a confirmatory HIV test, study staff will offer an HIV self-test kit prior to enrolment, to confirm a positive HIV status. Those who choose to initiate ART will receive the standard Determine and Unigold confirmatory tests prior to ART initiation, following routine care.

### Study outcomes

The primary outcome is the proportion of men who are retained in ART care 6 months after (re-)engagement. Secondary outcomes include: (1) ART initiation; (2) adverse events experienced by men or their female partners (ie, unwanted disclosure, end of relationship or IPV; (3) intervention acceptability and (4) cost-effectiveness. ART retention outcomes will be measured through medical chart reviews, while secondary outcomes will be measured through self-reports. Process outcomes include: (1) the proportion of men who were successfully traced; (2) the proportion of eligible men who consented to participate; (3) men's experience with the intervention and (4) the quality of the intervention.

### Sample size considerations

We powered the study to detect differences in 6-month retention between stepped and low-intensity arms, and the stepped and high-intensity arms. Based on pilot data, we assumed that 40% of men in the low-intensity arm, 60% in the stepped arm and 80% in the high-intensity arm will engage in ART and be retained at 6 months. Any man lost to follow-up will be treated as a failure for the outcome evaluation. With 181 men per arm and 20% loss to follow-up from the study in all arms, the power for detecting the specified differences between stepped and low-intensity arms and between stepped

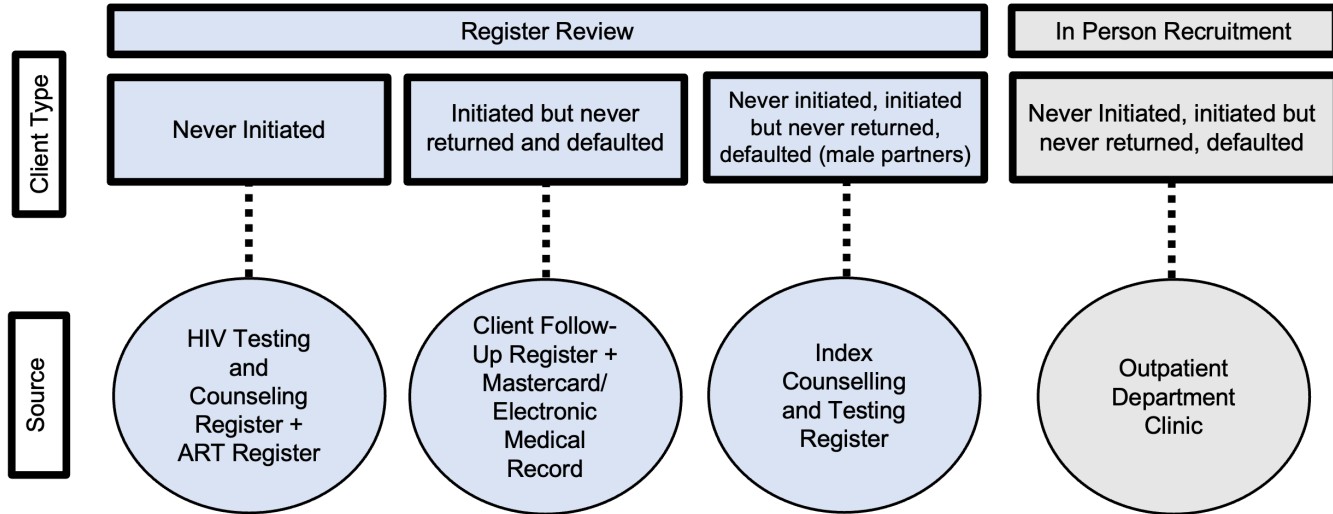

**Figure 2** Recruitment sources and ART disengagement criteria by recruitment type. ART, antiretroviral therapy.

and high-intensity arms will be 0.8, with test level 0.025 after Bonferroni adjustment for two comparisons. We will have 0.99 power to detect specified differences between low-intensity and high-intensity arms. The calculation is based on asymptotic normality of log OR.[41] We need to enrol and randomise 181 men per arm (a total of 543 men living with HIV).

### Data collection
Study recruitment, enrolment and data collection will be conducted by male study staff, who are distinct from local HCWs implementing the interventions.

### Recruitment
Men will be identified through both medical register reviews and in-person recruitment at participating health facilities. Various medical charts will be reviewed to identify different types of eligible men: HIV testing and counselling and ART registers to identify men who tested HIV positive but never initiated ART; client follow-up registers to identify those who initiated but never returned for their first ART appointment or those who defaulted from care; and index counselling and testing registers to identify male partners of female ART clients (figure 2). In-person recruitment will involve screening men at outpatient departments (OPDs) because our previous research has found that men in Malawi frequent OPD settings for health needs,[17] and our formative work suggests that men who disengage from ART services still frequent the OPD for care. In-person recruitment will be used for all client types.

### Tracing and eligibility screening
Study staff will trace potential participants identified through medical chart reviews via phone (if available) or home visits based on tracing data provided in medical documentation. All potential participants will be traced up to three times before being considered lost

to follow-up. All screening and enrolment processes will take place in person.

### Consent, enrolment and baseline survey
Men who are eligible for the study will complete written informed consent and complete a baseline survey immediately following enrolment. The baseline survey will collect data on key demographic variables (marital status, number of children, employment, self-rated health) and previous engagement with HIV and non-HIV health services. All surveys will be conducted in the local language (Chichewa) by trained study staff using electronic tablets. Surveys will be programmed using SurveyCTO software (http://www.surveycto.com).

### Follow-up data
Study staff will administer follow-up surveys at 2 and 4 months after enrolment. Follow-up surveys will measure exposure to (and acceptability of) the interventions, changes in key demographics since enrolment (ie, marital status, number of children, employment, self-rated health), any adverse events since enrolment (ie, unwanted status disclosure, termination of relationship due to the intervention) and use of ART services. The location and specific time of the follow-up survey will be based on participant preference.

Medical chart reviews will be conducted to assess men's engagement with ART services 6 months after study enrolment. Individuals without a medical chart outcome will be followed up in person and their health passport, a pocket medical record where providers record data during health visits, will be reviewed to collect the ART outcome. Men who cannot be reached or are lost to follow-up in any arm will be counted as failures for that specific ART outcome of interest: (re)initiation or 6-month retention.

## Patient and public involvement

Extensive formative work informed the development of the study protocol including in-depth interviews, focus group discussions and a systematic literature review. The study protocol and tools were presented to Ministry of Health, national stakeholders and implementing partners (see online supplemental file 2).

## Cost data

The average cost per successful outcome (6-month retention) will be calculated and compared across arms incrementally. We will use micro-costing methods by creating an inventory of the resources used to achieve the observed study outcomes including: (1) standard counselling interactions (staff cadre, training received, duration of interaction and distance from facility travelled where applicable); (2) motivational interviewing interactions (staff cadre, training received, duration of interaction and distance from facility travelled where applicable); (3) provider interactions (staff cadre, training received, duration of interaction and distance from facility travelled where applicable); and (4) cost of reminder messages sent, when messages are delivered telephonically instead of in person. For each study patient, the quantity (number of units) of resources used will be determined. Costs will be measured from the healthcare provider perspective. Unit costs of resources, which are not human subject data, will be obtained from external suppliers and the health facilities' finance and procurement records and multiplied by the resource usage data to provide an average cost per study patient in each study arm. A cost-effectiveness analysis will be conducted by dividing the incremental cost between two arms by the incremental effectiveness (number of people retained at 6 months) in the respective arms.

## Analysis plan

Data analysis will be conducted in R: A Language and Environment for Statistical Computing (R Foundation for Statistical Computing). We will use the Consolidated Standards of Reporting Trials standards for reporting trial outcomes.[42] Using an intention-to-treat analysis, all randomised men will be included in the analysis of the primary outcome; men with missing outcomes due to loss to follow-up will be treated as outcome failures. We will calculate descriptive statistics, including mean, SD, range and frequency distributions for the demographic characteristics and study outcomes by study arm. The primary outcome and all other binary outcomes will be analysed by logistic regression models with age, marital status, healthcare facility and other key sociodemographic variables included as covariates. The intervention effects will be tested by Wald tests of the relevant regression parameters. The hypotheses of no difference between the stepped arm and the low/high-intensity arm will be rejected if the p value is smaller than 0.025 (Bonferroni adjustment). CIs for ORs comparing the stepped arm with the low-intensity and high-intensity arms with coverage probabilities 0.975 will be calculated by profile likelihood methods. Due to the Bonferroni adjustment, the simultaneous coverage probability of both intervals will be at least 0.95. To address the secondary objectives, more elaborate logistic regression models will be built for each of the binary outcomes with available individual-level, community-level and facility-level factors included as covariates in addition to the intervention status.

## Nested studies

A series of nested, mixed-methods studies will be conducted to identify factors associated with ART engagement within each intervention arm, and to explore the implementation and acceptability of interventions.

### Qualitative data collection

Guided by Grounded Theory, we will conduct cross-sectional in-depth interviews with a random subset of 40 male participants per arm (120 total) throughout the study period. Clients will be randomly selected at various times of the study using computer-generated randomisation, stratifying the sample by arm and successful trial outcomes (ie, did clients reinitiate ART and/or reach 6-month retention). Data will assess characteristics of men who fail to engage in care, contextualise decisions around ART initiation and retention, and identify additional strategies that may be needed for men to successfully engage and be retained in ART programmes (see online supplemental file 3). Data collection tools and analysis plans will be informed by the Andersen's Emerging Model of Health Services Use, phase 4[43] that examines multilevel factors that influence health outcomes. Specifically, it examines the interaction of: (1) environment and structure of health services; (2) clients' enabling resources and (3) clients' perceived need/motivation to access services. Qualitatively understanding how the IDEaL interventions influence these levels, and what barriers still remain, will help refine future interventions.

Interviews will be conducted by a trained male interviewer in the local language. Interviews will be digitally recorded, transcribed and translated into English for analysis. Four investigators will pilot a codebook by independently reading and coding a randomly selected subset of transcripts. Through an iterative consultative process and using iterative and deductive coding strategies, each investigator will revise their respective codebook until there is high inter-rater reliability among the group. All transcripts will be coded in Atlas.ti V.8.3[44] and text analysed using constant comparison methods[45] to compare and contrast themes that arise within and between interventions and trial outcomes.

### Implementation log sheet

During the course of the intervention, HCWs will keep daily logs as one of the study monitoring and evaluation tools to assess the implementation of the intervention for each participant. Primary events to be recorded in the daily logs are: (1) unable to reach participant (and reason); (2) contacted participant; (3) intervention provided (and notes about the challenges and successes of the interaction) and (4) other comments relevant to intervention implementation. Each

event will be recorded with a corresponding date. Logs will be digitised in English. Findings may influence how similar interventions are implemented in the future.

## Ethics and dissemination

The IDEaL trial is registered with ClinicalTrials.gov as NCT05137210. The protocol was approved by the Institutional Review Board of the University of California, Los Angeles and the National Health Sciences Research Committee in Malawi. Study findings will be disseminated through peer-reviewed journal articles, national and international conference presentations, and meetings with Malawi Ministry of Health, facility and community stakeholders.

## DISCUSSION

Studies have reported poorer outcomes for HIV testing, treatment initiation and treatment adherence in men compared with women[2] for over a decade.[46 47] Men are often portrayed as difficult, hard-to-reach and actively avoiding health facilities. In IDEaL, we aim to investigate whether men really are hard to reach or will men engage in care when services are offered in ways that are accessible to them and resonate with their needs, as growing evidence suggests.[17] We propose testing a stepped intervention that increases in intensity over time against low-intensity and high-intensity interventions—all tailored to men—to identify the most cost-effective strategy to (re) initiate men in HIV treatment services in Malawi.

IDEaL is different from other ART engagement and re-engagement interventions in several important ways. First, we will enrol men living with HIV across the treatment cascade, including those who have never initiated ART, those who are at risk of immediate default after initiation and those who have been in care but subsequently defaulted. Formative research suggests that barriers to ART initiation and reinitiation may be similar[48]; however, most interventions focus specifically on either first-time initiation or re-engagement, but not both. Our study will assess if one overarching programme can improve men's engagement across the treatment cascade, regardless of whether they are starting ART for the first time or returning to care after a period of disengagement. One overarching intervention may be more scalable than multiple, separate interventions across the cascade. Second, we tailor interventions to men's unique needs and motivations, based on extensive formative work. While innovative interventions for men are underway,[25–27] few have rigorously tested the impact of male-tailored interventions on ART engagement.[49]

Finally, we will test a stepped intervention that builds in intensity over time until men (re)initiate care. This approach allows men who are ready to (re)initiate to do so at minimal cost to the health system, while those who need additional support can receive more resource-intensive interventions to support their ART engagement.[31] Stepped interventions have been effective in other settings and can address multiple barriers faced by the target population with minimal cost.[31 32] Findings from IDEaL will provide crucial knowledge of how best men can be reached and can inform intervention scale-up.

**Author affiliations**
[1]Division of Infectious Diseases, David Geffen School of Medicine, University of California Los Angeles, Los Angeles, California, USA
[2]Department of Implementation Science, Partners in Hope, Lilongwe, Malawi
[3]Department of Global Health, Boston University School of Public Health, Boston, Massachusetts, USA
[4]Department of Medical Microbiology, Amsterdam University Medical Center, University of Amsterdam, Amsterdam, Netherlands
[5]Foundation for Innovative New Diagnostics (FIND), Geneva, Switzerland
[6]Global Health Institute, University of California, San Francisco, California, USA
[7]Department of Probability and Statistics, Faculty of Mathematics and Physics, Charles University, Prague, Czechia
[8]Health Economics and Epidemiology Research Office, Department of Internal Medicine, School of Clinical Medicine, Faculty of Health Sciences, University of the Witwatersrand, Johannesburg, South Africa
[9]Clinical Research Programme, Malawi Liverpool Wellcome Programme, Blantyre, Malawi
[10]Liverpool School of Tropical Medicine, Liverpool, UK

**Acknowledgements** We wish to thank the Malawi Ministry of Health for their support of this trial. We would also like to acknowledge Joep van Oosterhout, Misheck Mphande, Isabella Robson, Thoko Banda, Peter Mwamlima and Eric Lungu for their contributions to protocol development and for their support of study implementation.

**Contributors** KD and AC conceptualised the study. KD is responsible for funding acquisition. KD, KB, JH, KP, BN, RH and AC developed study protocol and materials. JH, KB, KP and EC will implement the study. KD, MK, KB, TJC, BN, LCL, TJC and AC developed the analysis plan, and KD, MK, KB and AC will analyse the data. KD and EC wrote the first draft, and KB, JH, KP, BN, LCL, RH, SP, EC, RH, TJC and AC edited following drafts. All authors have read and approved the final manuscript.

**Funding** The work was supported by the Bill and Melinda Gates Foundation (grant number INV-001423). KD was supported by the National Institute of Mental Health of the National Institutes of Health (grant number R01-MH122308), Fogarty International (grant number K01-TW011484-01) and UCLA GSTTP (grant number N/A). LCL was supported by the National Institute of Mental Health of the National Institutes of Health under grant number K01MH119923.

**Disclaimer** The content is solely the responsibility of the authors and does not necessarily represent the official views of the National Institutes of Health.

**Competing interests** None declared.

**Patient and public involvement** Patients and/or the public were involved in the design, or conduct, or reporting, or dissemination plans of this research. Refer to the Methods section for further details.

**Patient consent for publication** Obtained.

**Provenance and peer review** Not commissioned; externally peer reviewed.

ORCID iDs
Kathryn Dovel http://orcid.org/0000-0002-5622-3401
Brooke E Nichols http://orcid.org/0000-0003-4682-4999

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
