## [Reviewer comments · BMJ Open]

ARTICLE DETAILS

TITLE (PROVISIONAL)	Identifying efficient linkage strategies for HIV self-testing (IDEaL): a study protocol for an individually randomized control trial
AUTHORS	Dovel , Kathryn; Balakasi, Kelvin; Hubbard, Julie; Phiri, Khumbo; Nichols, Brooke; Coates, Thomas; Kulich, Michal; Chikuse, Elijah; Phiri, Sam; Long, Lawrence; Hoffman, Risa; Choko, Augustine

VERSION 1 – REVIEW

REVIEWER	Mugenyi, Levi MRC/UVRI and LSHTM Uganda Research Unit, Statistics
REVIEW RETURNED	28-Jan-2023

GENERAL COMMENTS	This study aims to determine which one of the three proposed interventions is better at retaining men living with HIV on ART six months after initiating treatment (among other objectives). The justification for the study is very clear. I find the study very important towards eliminating HIV. I have a few comments below which I think that if addressed will help improve the protocol. 1. Abstract: While the primary and secondary outcomes are well stated, it's not clear what statistical method(s) the investigators will use to determine the difference (e.g., retention) between the study arms 2. Methods and analysis: a. Under interventions sub-section (page 6), the authors mention facility navigation when defining the different arms, however, it's not clear what they mean by "navigation". A sentence to define this in the paragraph it is first mentioned would be helpful. Also, it seems like the difference between Low- and High-intensity arms is that the latter includes outside-facility ART initiation, should we assume that for the Low-intensity and stepped arm we have only inside-facility ART initiation? Please clarify. Again, the High-intensity arm indicates that facility-navigation is for follow-up ART visits, but it's not clear what the facility-navigation for the Low-intensity arm entails. A distinction between intervention arms should be very clear. b. Under Arm 1: Stepped Arm (page 6), the authors state that individuals will move to the next step every 2-weeks moving from the lightest to the most intensive interventions over the course of X weeks. Can the authors clarify the number of weeks (the value of X)? If it is not feasible to tell the value of X a priori, can we have an idea of the possible values of X? c. Under Arm 1: Stepped Arm (page 8), the authors state that the nurse will schedule a 4-week follow-up ART refill appointment at the health facility of the man's choice. Will the health facilities be pre-listed to the client, or the client will have liberty to choose to go anywhere even if it's a very expensive facility? Who meets the related costs at the facility preferred by the client? Will these be only public facilities? if so, be clear by mentioning public facility
---

	everywhere. d. Under Arm 2: Low-Intensity Arm (page 8), the authors propose to have the intensive counselling session given by a low-level cadre male HIV counsellor. I think low-cadre definition may vary from setting to setting (or between countries). A clarification of who is considered a low cadre in the study setting would be helpful. e. Under Arm 2: Low-Intensity Arm (page 8). How will those who wish not to be contacted be analysed? It would be interesting to know which arm has more refusals. Will the secondary outcome "ART initiation" handle this? Please clarify. f. Study outcomes (page 10): For the secondary outcomes, probably for clients who refuse to join the trial, it would be nice to know proportion self-reporting or don't report ART initiation by arm. Would refusal be associated with self-report of ART initiation? g. Sample size considerations (page 10): The authors calculated a sample size of 181 per arm after accounting for 20% loss to follow-up with 80% power of detecting difference in retention between Low-intensity and Stepped arms assuming retention of 40% for the Low-intensity arm and 60% for the stepped arm. Using the same assumptions and applying the STATA power command "power twoproportions 0.4 0.6, n(100(10)400)" I noticed that power far exceeds 80% (not just 80% stated in the protocol) with the sample of 181. In fact you already attained at least 80% for n=100. Please check it. Also, I think it's important to account for facility-level clustering when calculating the sample size? You only accounted for loss-to-follow up. Probably retention will vary by facility where clients initiate ART because patients care may not be same across facilities. h. In the study design, the authors state this as a non-inferiority-controlled trial, however, the sample size calculation is silent about the non-inferiority design. For example, by what margin will the arms be concluded non-inferior? Probably the choice of the word non-inferiority in the design is misleading. i. Analysis plan (page 13): The authors state that missing outcome assessment due to loss to follow-up will be treated as outcome failures, however, it's not clear how these failures will be considered when estimating retention. Will you exclude them? Will you do imputation for missing outcome? Please clarify. Also, how will you account for other outcomes like death and transfer of care outside trial facilities? j. Analysis plan (page 13): The authors propose to use a logistic regression model, which is an appropriate method, but it's not clear how they will account for facility-level clustering where participants will be enrolled on ART. k. Analysis plan (page 13): The authors propose to use Bonferroni adjustment but they don't state what p-value or (other ways) they intend to conclude statistical significance after the adjustment. Also, I am not sure you need multiple comparisons because above (under intervention subsection) you state that you will compare Low-intensity arm to stepped arm, and High-intensity arm to stepped arm. The same approach was used for sample size calculation, meaning there is no multiple comparisons. Please explain why you need Bonferroni adjustment?
--	---

REVIEWER	Regencia, Zypher Jude University of the Philippines Manila
REVIEW RETURNED	29-Jan-2023

GENERAL COMMENTS	This is a very interesting protocol manuscript tackling a very important public health problem. I have some issues/concerns:
--

Introduction:

1. Line 31, page 4: Cite the reference
2. Lines 36 to 40, page 4: The sentence needs references to substantiate the claims.
3. Kindly provide more substantiable evidence to prove that there are non-engaged men in SSA by citing or estimating the numbers of these non-engaged men in SSA. What is the current rate of HIV infection in SSA? And how many of the men infected with HIV are currently on ART? How many are not? The answers to these questions may help solidify the statement of the problem.
4. Kindly revise the introduction to enrich the gaps identified and how will your research be also to help minimizing or eliminating the public health problem identified.

Methods

1. For the objectives, the word “impact” may be a bit far-fetched since “impact” implies a long-term effect. Kindly replace it with “effect”.
2. The authors mentioned that: Individual men will be block randomized by a biostatistician using a 1:1:1 ratio to either the Stepped, Low-Intensity, or High-Intensity study arm using a computer-generated program. What is this software or program to be used? Kindly specify.
3. The authors mentioned “counseling curriculum developed specifically for this trial”. Were the results of this curriculum development published? If yes, kindly cite so that we may also review the content.
4. The authors mentioned “validated form of counseling for motivational.” Is this form still valid since this was published in 1995. Or are there any other current or updated forms you may use? Also, will this form be adapted since this form is generally for counseling and not specific for ART.
5. The authors mentioned “based on previous trials...” in line 29, page 11. What are these trials? Kindly cite the references.
6. The authors mentioned that a male nurse will be involved in Step 3: Outside-facility ART initiation + male-specific counseling + facility navigation. Is the male nurse also a certified HIV counselor/mentor?
7. Kindly record also the “refusal rate” as this is an important data. Reasons for refusal may also be recorded as in-person recruitment will be done. This is for future researches.
8. The authors mentioned in the objectives that these are the secondary outcome: ART initiation; (2) the presence of adverse events (i.e., unwanted disclosure, end of relationship, or intimate partner violence (IPV)); (3) intervention acceptability; and (4) cost-effectiveness. However, in the data collection guidelines, there are explicit ways elaborated to

	measure the ART initiation, presence of adverse events, and the intervention acceptability. There should be a section on how to achieve these secondary outcomes.  9. For the qualitative study, what design will be used? 10. Kindly justify the use of 120 male participants for the qualitative data collection. 120 informants are too many and exhaustive in terms of labor in transcribing. Will data saturation be observed? 11. How will you select the participants in the qualitative data collection? Will they be randomized? If purposive sampling will be used, what are the criteria to be eligible? 12. For the qualitative data collection, kindly attach as supplemental the guide to in-depth interviews and the guide questions. 13. For the qualitative data collection, what type of analysis will be used? How many will be involved to attain consensus? 14. What language will be used for the qualitative data collection? Discussion  1. Lines 46-48, page 14: Cite the reference/s. 2. The authors mentioned that they will measure female partner perceptions of the feasibility and acceptability of a male-only intervention. This was not included in the methods section. 3. Include the limitations of this research protocol. What are the possible mishaps that may happen during the implementation? What will be the barriers towards the successful implementation? What are the scopes that your research only addresses and what are not included in these scopes? Enrich the discussion section. Others  1. The Strengths and Limitations section only provided strengths and did not mention any limitations. 2. The authors will implement a qualitative data collection. However, they did not mention if the qualitative findings will be triangulated with the quantitative findings. Kindly put a section on how will the data be triangulated and what type of approach to be used. 3. The manuscript lacks an ethics section elaborating the ways on how to protect confidentiality and privacy of the participants. Data management plan is also not well addressed.
--	--

REVIEWER	Shrivastava, Saurabh R. Shri Sathya Sai Med Coll
-----------------	---

REVIEW RETURNED	02-Feb-2023
GENERAL COMMENTS	An interesting RCT to improve the uptake of ART among men. Suggestions for improvement:  1. Specify study variables 2. Mention various sub-group analysis that have been carried out 3. Check for the chronology of references (In the introduction section) 4. How Bias Minimization and Bias ascertainment was done in the RCT?

VERSION 1 – AUTHOR RESPONSE

Reviewer #1

1. Abstract: While the primary and secondary outcomes are well stated, it's not clear what statistical method(s) the investigators will use to determine the difference (e.g., retention) between the study arms

We have added our statistical methods in the abstract as follows: "The primary analysis will be intention to treat with all eligible men in the denominator and all men retain in care at 6 months in the numerator. The proportions achieving the primary outcome will then be compared with a risk ratio, corresponding 95% confidence interval and p-value computed using binomial regression while accounting for clustering at facility level."

2. Methods and analysis:

a. Under interventions sub-section (page 6), the authors mention facility navigation when defining the different arms, however, it's not clear what they mean by "navigation". A sentence to define this in the paragraph it is first mentioned would be helpful.

Thank you for this comment. We have added a definition for facility navigation in the Methods and Analysis section.

Also, it seems like the difference between Low- and High-intensity arms is that the latter includes outside-facility ART initiation, should we assume that for the Low-intensity and stepped arm we have only inside-facility ART initiation? Please clarify.

This is clarified in the Methods and Analysis section. Briefly, the Stepped arm offers outside-facility ART initiation only for men who do not (re-)initiate ART within the first 4 weeks after study enrollment. The first step of the Arm is exactly the same as the Low-Intensity Arm. Second step adds psychosocial, ongoing counseling, and third step (for those still not re-engaged in care) is outside-facility ART services.

Again, the High-intensity arm indicates that facility-navigation is for follow-up ART visits, but it's not clear what the facility-navigation for the Low-intensity arm entails. A distinction between intervention arms should be very clear.

Thank you for raising this point. Per the earlier comment, we have now provided a definition for facility navigation in the Methods and Analysis section. Facility navigation activities is the same across arms. The only difference is when facility navigation takes place.

b. Under Arm 1: Stepped Arm (page 6), the authors state that individuals will move to the next step every 2-weeks moving from the lightest to the most intensive interventions over the course of X weeks. Can the authors clarify the number of weeks (the value of X)? If it is not feasible to tell the value of X a priori, can we have an idea of the possible values of X?

Thank you for this catch, this was an editing error. The X has been removed

c. Under Arm 1: Stepped Arm (page 8), the authors state that the nurse will schedule a 4-week follow-up ART refill appointment at the health facility of the man's choice. Will the health facilities be pre-

listed to the client, or the client will have liberty to choose to go anywhere even if it's a very expensive facility? Who meets the related costs at the facility preferred by the client? Will these be only public facilities? if so, be clear by mentioning public facility everywhere.

Men can choose to access care anywhere, per Malawian guidelines. However, the study will not cover costs incurred for travel or accessing services. This is now clarified. Those who are WHO Stage 3 or 4 and require additional immediate services will be referred to the nearest public hospital, which is free of charge. The vast majority of HIV care in Malawi is public, or when private, does not require payment. Based on our extensive work in facilities throughout Malawi, we are confident that offering men their choice of public facility will not limit men's choices, as private clinics are so rare.

To address costs related to seeking services we have added the following statement (Methods and Analysis section) "The client will be responsible for all transport costs related to return to the facility in all study arms." We agree that this is a necessary point of clarification as cost is an important point that could explain loss from care and could impact the primary outcome.

d. Under Arm 2: Low-Intensity Arm (page 8), the authors propose to have the intensive counselling session given by a low-level cadre male HIV counsellor. I think low-cadre definition may vary from setting to setting (or between countries). A clarification of who is considered a low cadre in the study setting would be helpful.

We have adjusted this language to 'lay cadre' and provided the following definition in the Methods and Analysis section: "Patient supporters are responsible for routine tracing, linkage support medical record documentation and counseling."

e. Under Arm 2: Low-Intensity Arm (page 8). How will those who wish not to be contacted be analysed? It would be interesting to know which arm has more refusals. Will the secondary outcome "ART initiation" handle this? Please clarify.

We will report the numbers and proportions of those who could not be contacted in each arm. In the primary analysis, they will be included as treatment failures, that is, not retained in ART care. It is unlikely that men who cannot be contacted continue ART elsewhere. Thus, this is a most sensible way to handle dropouts in the primary outcome. In secondary analyses, they be included if at least partial information on the outcome is available.

f. Study outcomes (page 10): For the secondary outcomes, probably for clients who refuse to join the trial, it would be nice to know proportion self-reporting or don't report ART initiation by arm. Would refusal be associated with self-report of ART initiation?

We will have information about past ART initiation and many other variables in the screening data. The characteristics of those who refused participation in the trial will be reported and compared with the enrolled participants.

g. Sample size considerations (page 10): The authors calculated a sample size of 181 per arm after accounting for 20% loss to follow-up with 80% power of detecting difference in retention between Low-intensity and Stepped arms assuming retention of 40% for the Low-intensity arm and 60% for the stepped arm. Using the same assumptions and applying the STATA power command "power two proportions 0.4 0.6, n(100(10)400)" I noticed that power far exceeds 80% (not just 80% stated in the protocol) with the sample of 181. In fact you already attained at least 80% for n=100. Please check it. Also, I think it's important to account for facility-level clustering when calculating the sample size? You only accounted for loss-to-follow up. Probably retention will vary by facility where clients initiate ART because patients care may not be same across facilities.

There were two reasons why we arrived at a sample size much larger than 100 (which would indeed suffice in the circumstances described in the comment). First, we counted the expected 20% of subjects who would be impossible to contact as failures – thus the resulting retention proportions were lower (0.32 and 0.48, respectively). Second, we calculated the sample size for alpha level $0.05/2=0.025$ to adjust for performing two tests. Facility-level clustering was not taken into account in the sample size calculation because this was not a cluster-randomized trial and randomization was stratified by facility. Facility effects will be taken into account in the final analysis and the actual power could not be worse than what the simplified sample size calculation indicated.

h. In the study design, the authors state this as a non-inferiority-controlled trial, however, the sample size calculation is silent about the non-inferiority design. For example, by what margin will the arms be concluded non-inferior? Probably the choice of the word non-inferiority in the design is misleading.

Thank you for pointing out this inconsistency. The trial should not be described as a non-inferiority trial. This has been removed in the manuscript.

i. Analysis plan (page 13): The authors state that missing outcome assessment due to loss to follow-up will be treated as outcome failures, however, it's not clear how these failures will be considered when estimating retention. Will you exclude them? Will you do imputation for missing outcome? Please clarify. Also, how will you account for other outcomes like death and transfer of care outside trial facilities?

Subjects who will be lost to follow-up will be counted as if they were not retained in ART care. Most of them will indeed be out of care rather than secretly transferring to an external facility and continuing in care without the study staff knowing it, so this solution is sensible. We have clarified this in the revision. The deaths (most likely few, if any) will be treated as outcome failures as well. Subjects who transferred to other facilities will be traced (if a contact can be made) and their outcome will be evaluated.

j. Analysis plan (page 13): The authors propose to use a logistic regression model, which is an appropriate method, but it's not clear how they will account for facility-level clustering where participants will be enrolled on ART.

Facility will be used as a categorical variable in the logistic regression model, thus allowing different facilities to have different retention probabilities. We considered facility among the "key sociodemographic variables" to be included as covariates. In the revision, we mention facility explicitly.

k. Analysis plan (page 13): The authors propose to use Bonferroni adjustment but they don't state what p-value or (other ways) they intend to conclude statistical significance after the adjustment. Also, I am not sure you need multiple comparisons because above (under intervention subsection) you state that you will compare Low-intensity arm to stepped arm, and High-intensity arm to stepped arm. The same approach was used for sample size calculation, meaning there is no multiple comparisons. Please explain why you need Bonferroni adjustment?

Bonferroni adjustment is needed to keep an overall control over the Type I error for both comparisons: Low-intensity arm to Stepped arm, and High-intensity arm to Stepped arm. We will perform two tests and Bonferroni adjustment will assure that the probability of rejecting at least one hypothesis when both are actually correct is not more than 0.05. The same applies to the confidence intervals (which are even more important for the interpretation of the study results). We will report a confidence interval for the OR comparing the Stepped arm to the Low-intensity arm and another confidence interval for the OR comparing the Stepped arm to the High-intensity arm. Bonferroni adjustment assures that both confidence intervals cover the true OR's with a combined probability of at least 0.95.

Reviewer #2

Introduction:

1. Line 31, page 4: Cite the reference

2. Lines 36 to 40, page 4: The sentence needs references to substantiate the claims.

We have added additional citations throughout

3. Kindly provide more substantial evidence to prove that there are non-engaged men in SSA by citing or estimating the numbers of these non-engaged men in SSA. What is the current rate of HIV infection in SSA? And how many of the men infected with HIV are currently on ART? How many are not? The answers to these questions may help solidify the statement of the problem.

Thank you for this. We have strengthened the introduction to include more evidence regarding the problem, and the gap our trial addresses.

4. Kindly revise the introduction to enrich the gaps identified and how will your research be also to help minimizing or eliminating the public health problem identified.

Additional justification has been added at the end of the introduction.

Methods

5. For the objectives, the word "impact" may be a bit far-fetched since "impact" implies a long-term effect. Kindly replace it with "effect".

This has been changed to 'effect'

6. The authors mentioned that: Individual men will be block randomized by a biostatistician using a 1:1:1 ratio to either the Stepped, Low-Intensity, or High-Intensity study arm using a computer-generated program. What is this software or program to be used? Kindly specify.
The software used for randomization is R. We have clarified this in the text in the Methods and Analysis section, paragraph 3.

7. The authors mentioned "counseling curriculum developed specifically for this trial". Were the results of this curriculum development published? If yes, kindly cite so that we may also review the content. We have two manuscripts in progress for the male-specific counseling developed for the trial: one to present the process of curriculum development for male-specific counseling, the second to present a quality assessment during its early piloting and implementation. However, they have not yet been published. We look forward to sharing this work as soon as it is published.

8. The authors mentioned "validated form of counseling for motivational." Is this form still valid since this was published in 1995. Or are there any other current or updated forms you may use? Also, will this form be adapted since this form is generally for counseling and not specific for ART.
Thank you for this clarification. We have removed the word "validated" and now simply say "Motivational interviewing is a client-centered, client-led method for counseling that helps participants identify barriers to a desired outcome and develop personalized solutions". Motivational interviewing has previously been successful with ART clients. We have updated citations and include citations on motivational interviewing among ART clients.

9. The authors mentioned "based on previous trials..." in line 29, page 11. What are these trials? Kindly cite the references.
This text is incorrect. It should have said, "based on trial pilot data", and is now corrected. No citation available.

10. The authors mentioned that a male nurse will be involved in Step 3: Outside-facility ART initiation + male-specific counseling + facility navigation. Is the male nurse also a certified HIV counselor/mentor?
All study nurses are certified ART providers trained in HIV testing, counseling, and treatment. This clarification has been made in the Methods and Analysis section.

11. Kindly record also the "refusal rate" as this is an important data. Reasons for refusal may also be recorded as in-person recruitment will be done. This is for future researches.
This data will be collected and reported. Further description of how refusals will be addressed are above in the responses to Reviewer 1 (questions e and f).

12. The authors mentioned in the objectives that these are the secondary outcome: ART initiation; (2) the presence of adverse events (i.e., unwanted disclosure, end of relationship, or intimate partner violence (IPV)); (3) intervention acceptability; and (4) cost-effectiveness. However, in the data collection guidelines, there are explicit ways elaborated to measure the ART initiation, presence of adverse events, and the intervention acceptability. There should be a section on how to achieve these secondary outcomes.
Data collection related to the secondary outcomes are clearly outlined in the 'Follow-up Data' and 'Qualitative data section' sections in Methods and Analysis. ART initiation will be assessed using the medical chart review, and adverse events and intervention acceptability will be assessed by the 2- and 4- month follow-up surveys. We feel these are sufficiently addressed in the text as is and do not warrant their own section.

9. For the qualitative study, what design will be used?
The following text is added to the manuscript: Data collection tools and analysis plans will be informed by the Andersen's Emerging Model of Health Services Use, phase 4 (Anderson 2008) that examines multi-level factors that influence health outcomes. Specifically, it examines the interaction of: 1) environment and structure of health services; 2) clients' enabling resources; and 3) clients' perceived need/motivation to access services. Qualitatively understanding how the IDEaL interventions influence these levels, and what barriers still remain, will help refine future interventions.

10. Kindly justify the use of 120 male participants for the qualitative data collection. 120 informants are too many and exhaustive in terms of labor in transcribing. Will data saturation be observed? We will conduct in-depth interviews with a random subset of 40 male participants per arm (120 total) throughout the study period. We will stratify the sample by those who succeed in the trial (retained at 6 months) and those who do not (never (re-)initiated or not retained at 6 months). That means a total of 25 men in each category (ie. By arm and trial success). This is clarified in the manuscript.

In general, data should be collected until saturation is reached, meaning that no new themes or relevant information is emerging. The exact number of interviews required to reach saturation differs based on the aim of the study, the diversity in respondents, and the theoretical framework used for analysis. However, a basic rule of thumb is that no sample size should be under 25 participants in order to reach saturation and identify all relevant themes or new information important to the study (see citation below). Our research department has extensive experience in qualitative interview transcription, coding and analysis and are confident the 120 interviews are feasible and needed. Dworkin SL. Sample size policy for qualitative studies using in-depth interviews. In: Springer; 2012.

11. How will you select the participants in the qualitative data collection? Will they be randomized? If purposive sampling will be used, what are the criteria to be eligible?

The following text is now to the manuscript: Clients will be randomly selected at various times of the study using computer-generated randomization, stratifying the sample by arm and successful trial outcomes (i.e., did clients re-initiate ART and/or reach 6-month retention).

12. For the qualitative data collection, kindly attach as supplemental the guide to in-depth interviews and the guide questions.

Thank you for this suggestion. We will submit as a supplement.

13. For the qualitative data collection, what type of analysis will be used? How many will be involved to attain consensus?

Qualitative data will be analyzed using constant comparison, a facet of grounded theory. This is specified in the qualitative data collection section.

14. What language will be used for the qualitative data collection?

Interviews will be conducted in Chichewa, the local language. This has been specified in the Qualitative data collection section.

Discussion

1. Lines 46-48, page 14: Cite the reference/s.

Additional citations are added throughout.

2. Include the limitations of this research protocol. What are the possible mishaps that may happen during the implementation? What will be the barriers towards the successful implementation? What are the scopes that your research only addresses and what are not included in these scopes? Enrich the discussion section.

According to the manuscript guidelines, strengths and limitations were to be listed in the 'Strengths and Limitations' section following the abstract. We have now added a limitation to the Article Summary section, bullet point 4 "IDEaL does not address facility characteristics that may serve as barriers to men's use of facility-based services. Changing facility-based ART services requires a high level of resources and time in order to strengthen local infrastructure and change deeply entrenched cultural factors around the female-focused delivery of health services. While this is critical, immediate strategies are needed in the interim as more structural changes are developed."

Others

1. The Strengths and Limitations section only provided strengths and did not mention any limitations. Response provided in Discussion section, question 3 above.

2. The authors will implement a qualitative data collection. However, they did not mention if the qualitative findings will be triangulated with the quantitative findings. Kindly put a section on how will the data be triangulated and what type of approach to be used. Qualitative data will enrich quantitative data by providing in-depth understanding about why some clients succeed or fail to engage in care, and differences by arm. We will use quantitative data to stratify randomly selective participants for in-depth interviews, but we will not explicitly have a mixed methods analysis, combining and triangulating quantitative and qualitative data.

3. The manuscript lacks an ethics section elaborating the ways on how to protect confidentiality and privacy of the participants. Data management plan is also not well addressed.

Examining other BMJ Open Protocol papers, it is not common practice to have a detailed data management plan. Based on this, we have not elaborated on these topics within the manuscript, however, details regarding data management and confidentiality/privacy can be found in the clinical trials database.

Reviewer #3

1. Specify study variables

The demographic and HIV engagement variables collected during baseline are outlined in the Methods and Analysis section, paragraph 21. Primary outcome data collection in medical charts is outlined in the Methods and Analysis section, paragraph 22.

2. Mention various sub-group analysis that have been carried out

Subgroup analyses as such will not be performed but secondary analyses will consider interactions between the intervention arms and variables that have an important effect on the study outcomes. This will provide the desired information about how the interventions perform in various subgroups.

3. Check for the chronology of references (In the introduction section)

Thank you for this comment. Citations have been corrected.

4. How Bias Minimization and Bias ascertainment was done in the RCT?

We will explore balance on participant characteristics and account using multiple binomial regression for any imbalances observed

VERSION 2 – REVIEW

REVIEWER	Mugenyi, Levi MRC/UVRU and LSHTM Uganda Research Unit, Statistics
REVIEW RETURNED	18-Apr-2023

GENERAL COMMENTS	All my comments and concerns were well addressed, except one minor clarification (which does not require sending back the manuscript to me for review) 1. In the analysis plan, I raised a concern for accounting for facility level clustering in the analysis. The authors say this will be included as a categorical variable in the model. This is still not clear to me how the authors plan to include facility as categorical variable, I think there will be many facilities (than what defines a categorical variable) that participants will choose because these are not fixed a prior. I am imagining how many levels this categorical variable will have!! Will facilities be re-grouped in some sensible categories?
---

REVIEWER	Regencia, Zypher Jude University of the Philippines Manila
REVIEW RETURNED	30-Mar-2023

GENERAL COMMENTS	The authors did a good job in revising the manuscript and is now a better version from their previous one. I only have two comments: 1. The authors still did not mention the design for the qualitative part of the study. They only mentioned that they will follow the Andersen's Emerging Model of Health Services Use. 2. Kindly clarify how many investigators will be involved in the qualitative data analysis.
---

VERSION 2 – AUTHOR RESPONSE

Reviewer #1 - Dr. Levi Mugenyi, MRC/UVRI and LSHTM Uganda Research Unit

All my comments and concerns were well addressed, except one minor clarification (which does not require sending back the manuscript to me for review)

1. In the analysis plan, I raised a concern for accounting for facility level clustering in the analysis. The authors say this will be included as a categorical variable in the model. This is still not clear to me how the authors plan to include facility as categorical variable, I think there will be many facilities (than what defines a categorical variable) that participants will choose because these are not fixed a priori. I am imagining how many levels this categorical variable will have!! Will facilities be re-grouped in some sensible categories?

The participants of the IDEaL trial will receive services in one of 13 facilities. Thus, it is feasible to use facility as a categorical variable. There will be enough participants in each facility to estimate facility effects reliably. The number of facilities is provided in the Methods and Analysis, 'Population' section.

Reviewer #2 - Mr. Zypher Jude Regencia, University of the Philippines Manila

The authors did a good job in revising the manuscript and is now a better version from the previous one. I only have two comments:

1. The authors still did not mention the design for the qualitative part of the study. They only mentioned that they will follow the Andersen's Emerging Model of Health Services Use.

We will conduct one-on-one qualitative interviews using a Grounded Theory approach. This detail has been added to the manuscript (line 306).

2. Kindly clarify how many investigators will be involved in the qualitative data analysis.

Four investigators who specialize in qualitative analysis will take part in data analysis. This detail has been added to the manuscript (line 320).

VERSION 3 – REVIEW

REVIEWER	Regencia, Zypher Jude University of the Philippines Manila
REVIEW RETURNED	08-May-2023
GENERAL COMMENTS	The authors did a great job.